# Regulatory T Cell as a Biomarker of Treatment-Free Remission in Patients with Chronic Myeloid Leukemia

**DOI:** 10.3390/cancers13235904

**Published:** 2021-11-24

**Authors:** Yuki Fujioka, Daisuke Sugiyama, Itaru Matsumura, Yosuke Minami, Masatomo Miura, Yoshiko Atsuta, Shigeki Ohtake, Hitoshi Kiyoi, Yasushi Miyazaki, Hiroyoshi Nishikawa, Naoto Takahashi

**Affiliations:** 1Department of Hematology, Nephrology and Rheumatology, Akita University Graduate School of Medicine, 1-1-1 Hondo, Akita 010-8543, Japan; 2Division of Cancer Immunology, Exploratory Oncology Research and Clinical Trial Center (EPOC), National Cancer Center, Chiba 277-8577, Japan; hnishika@ncc.go.jp; 3Department of Immunology, Nagoya University Graduate School of Medicine, Nagoya 466-8550, Japan; dsugiyama@med.nagoya-u.ac.jp; 4Department of Hematology and Oncology, Kinki University Hospital, Osaka 589-8511, Japan; i.matsu@med.kindai.ac.jp; 5Department of Hematology, National Cancer Center Hospital East, Kashiwa 277-0882, Japan; yominami@east.ncc.go.jp; 6Department of Pharmacy, Akita University Hospital, Akita 010-8543, Japan; m-miura@hos.akita-u.ac.jp; 7The Japanese Data Center for Hematopoietic Cell Transplantation, Nagoya 461-0047, Japan; y-atsuta@jdchct.or.jp; 8Kanazawa University, Kanazawa 920-1192, Japan; sohtake@staff.kanazawa-u.ac.jp; 9Department of Hematology and Oncology, Nagoya University, Nagoya 464-8601, Japan; kiyoi@med.nagoya-u.ac.jp; 10Department of Hematology, Nagasaki University, Nagasaki 852-8521, Japan; y-miyaza@nagasaki-u.ac.jp; 11Division of Cancer Immunology, Research Institute, National Cancer Center, Tokyo 104-0045, Japan

**Keywords:** chronic myeloid leukemia, imatinib, treatment-free remission, regulatory T cells

## Abstract

**Simple Summary:**

Tyrosine kinase inhibitors (TKIs) have dramatically improved the treatment of chronic myeloid leukemia (CML). Recently, TKIs were discontinued in patients with CML with deep molecular remission, and some patients have been reported to be able to maintain long-term treatment-free remission (TFR). However, there is no certainty regarding which patients can maintain TFR. We focused on immunity in the TFR phase and investigated the immunological mechanism of continuous TFR or recurrence. Our results suggest that the group that maintains the TFR is immunologically activated. In addition, regulatory T cells can be used as a biomarker. These results may have important implications for future strategies for maintaining TFR in CML treatment.

**Abstract:**

Treatment-free remission (TFR) has become a therapeutic goal in chronic myeloid leukemia (CML), and approximately half of the patients with chronic phase-CML (CML-CP) with deep molecular remission (DMR) by tyrosine-kinase inhibitors (TKIs) have achieved TFR. However, the mechanism of continuous TFR is still unclear, as there are “fluctuate” patients who have BCR–ABL-positive leukemia cells but do not observe obvious relapse. We focused on the immune response and conducted an immune analysis using clinical samples from the imatinib discontinuation study, JALSG-STIM213. The results showed that, in the group that maintained TFR for 3 years, changes in regulatory T (Treg) cells were observed early after stopping imatinib treatment. The effector Treg (eTreg) cells increased transiently at 1 month after stopping imatinib and then returned to baseline at 3 months after stopping imatinib treatment. There was no difference in the Treg phenotype, and CD8^+^ T cells in the TFR group were relatively activated. High concentrations of imatinib before stopping were negatively correlated with eTreg cells after stopping imatinib. These data suggest immunological involvement in the maintenance of the TFR, and that Treg cells after stopping imatinib might be a biomarker for TFR. Furthermore, high imatinib exposure may have a negative immunological impact on the continuous TFR.

## 1. Introduction

Chronic myeloid leukemia (CML) occurs in hematopoietic stem cells and is characterized by the presence of the *BCR–ABL1* fusion gene on the Philadelphia chromosome. The advent of imatinib, the first *BCR–ABL1* tyrosine kinase inhibitor (TKI), has dramatically improved the prognosis of CML [1,2,3] and has become standard therapy for CML since 2001 [4,5]. Treatment-free remission (TFR) has become the new goal of CML therapy according to various TKI discontinuation trials [6,7,8,9,10,11]. The depth of remission, duration of TKI treatment, and deep molecular remission (DMR) are provided as prerequisites for a challenge to TFR in several guidelines; however, there are no corroborating biomarkers for successful TFR, and a detailed mechanism for maintaining TFR is currently unclear. TFR cannot be determined by estimating the absence or presence of *BCR–ABL1*-positive leukemia cells in the peripheral blood, because international-scale polymerase chain reaction (IS-PCR) has shown that patients with fluctuating values of the *BCR–ABL1* fusion gene can maintain TFR [10,11].

Although immune dysfunction is observed in patients with CML at diagnosis, immunological status is recovered during TKI treatment [12,13,14,15]. Imatinib reversibly impairs T-cell functions containing regulatory T (Treg) cells; however, it simultaneously induces the expansion of CD8 cytotoxic T lymphocytes (CTLs) and other immune cells [12,13]. It is believed that expanded CD8 CTLs carry out immunological elimination of leukemia cells. Several reports have referred to immune responses in TKI discontinuation trials; however, there are not yet any unified opinions [16,17,18,19,20,21]. Several reports have shown a correlation between an increase in natural killer (NK) cells at any timepoint and successful TFR [16,17,18,19].

In this study, we analyzed the immune profiles of patients with CML in an imatinib discontinuation trial. As immune conditions present large differences among individuals, especially in TKI-treated patients with CML, we focused on the changes in immune status in these patients before and after stopping imatinib administration. Mass cytometric screening analysis suggested alterations in Treg cells, and we investigated Treg fractionations in detail using flow cytometry (FCM). To analyze Treg cells, we classified the FoxP3-positive fractions into three categories: naïve Treg (nTreg) cells (CD45RA^+^FoxP3^low^CD4^+^), effector Treg (eTreg) cells (CD45RA^−^FoxP3^high^CD4^+^), and non-Treg cells (CD45RA^−^FoxP3^low^CD4^+^) [22,23,24]. We observed changes in eTreg cells and found that these changes could be useful as biomarkers for the prognosis of TFR. In addition, we found that the activation of CD8^+^ T cells as effector cells was also required for the eTreg cells. Sequential sampling from the same patients before and after stopping imatinib unrevealed the association between host immune status, influenced by imatinib exposure, and the maintenance of successful TFR after discontinuation of imatinib.

## 2. Materials and Methods

### 2.1. Patients and Samples

Peripheral blood samples were obtained from 68 patients participating in the JALSG-STIM213 trial, which was previously reported [11] according to the protocols approved by the ethics committee of Akita University and the National Cancer Center. The sampling timepoints were the same as those for the blood collection for the IS-PCR. All healthy individuals and patients with CML provided written informed consent before sampling according to the Declaration of Helsinki. Peripheral blood mononuclear cells (PBMCs) were isolated from heparinized blood by density gradient using Ficoll–Paque (GE Healthcare, IL, USA). PBMCs were directly subjected to ex vivo staining to analyze FoxP3^+^CD4^+^ T cells and the activation status of CD8^+^ T cells. Molecular responses were assessed regularly using IS-PCR every month for the first 6 months of the TFR phase, every 2 months for the next 6 months, and every 3 months thereafter. 

### 2.2. Mass Cytometry

Mass cytometry staining and analyses were performed as described previously [25]. The antibodies used for the mass cytometry analysis are summarized in Appendix A. PBMCs were stained after washing with phosphate-buffered saline (PBS) supplemented with 2% fetal calf serum (FCS). Cells were incubated in 5 μM Cell-ID Cisplatin solution (Fluidigm, CA, USA) in PBS, washed with MaxPar Cell Staining Buffer (Fluidigm), and stained with a mixture of surface antibodies. After washing, the cells were fixed and permeabilized using the Foxp3/Transcription Factor Staining Buffer Set (Thermo Fisher Scientific, MA, USA), according to the manufacturer’s instructions. The fixed and permeabilized cells were stained with intracellular antibodies. After washing twice, cells were rested in 125 nM MaxPar Intercalator-Ir (Fluidigm) diluted with 2% paraformaldehyde in PBS at 4 °C for 60 min. Cells were then washed twice with MaxPar Cell Staining Buffer and once with MaxPar water (Fluidigm), distilled water with minimal heavy-element contamination to reduce the background level. Cells suspended in MaxPar water supplemented with 10% EQ Four Element Calibration Beads (Fluidigm) were applied to the Helios system (Fluidigm), and data were acquired at a speed below 300 events/s. Data analyses were performed using FlowJo v10.7 (BD Biosciences, Franklin Lakes, CA, USA) as described previously [26].

### 2.3. Flow Cytometry (FCM)

FCM staining and analyses were performed as described previously [13,23,27]. The cell number for staining was 1–10 × 10^6^ cells. The antibodies used in the FCM analysis are summarized in Appendix A. PBMCs were washed with PBS supplemented with 2% FCS (washing solution) and stained with surface antibodies (1:20 to 1:50 dilution). Intracellular staining was performed with Foxp3/Transcription Factor Staining Buffer Set (Thermo Fisher Scientific, Waltham, MA, USA), according to the manufacturer’s instructions. After washing, the cells were analyzed using LSRFortessa (configured with 488, 640, 561, and 405 nm lasers, BD Biosciences, Franklin Lakes, CA, USA) and FlowJo software (BD Biosciences). The gating strategy of the FCM analysis is shown in Appendix A.

### 2.4. ELISA

Serum cytokine levels were quantified using the V-PLEX Proinflammatory Human Kit and U-PLEX TGF-β Human Kit (Meso Scale Diagnostics, Rockville, MD, USA). All assays were performed according to the manufacturer’s recommendations, using an alternative protocol. Cytokine concentrations were calculated using the Meso Scale Diagnostics Discovery Workbench software. Values outside the kit detection limits are not reported.

### 2.5. Imatinib Concentration

Peripheral blood was collected by venipuncture at 24 h (±2 h) after the oral administration of imatinib. Plasma was isolated by centrifugation at 1900× *g* for 15 min and stored at −40 °C until analysis. Imatinib trough concentrations were determined using high-performance liquid chromatography, as previously reported by us [28].

### 2.6. Cell Culture

PBMCs from healthy individuals were cultured in the presence of 10 IU/mL interleukin-2 (IL-2) and 20 mg/mL IL-7 with or without imatinib (CTL). Imatinib concentration was determined according to our previous report [13]. Imatinib was washed out 3 days later. Subsequently, PBMCs were stimulated with 0.5 µg/mL monoclonal antibody to CD3 (BioLegend, San Diego, CA, USA). PBMCs were sampled on days 0, 3, 4, 6, and 8 and subjected to FCM. 

### 2.7. Statistical Analysis

Prism 9 software (GraphPad) was used for the statistical analyses of our results. Patient characteristics were compared between the two populations—TFR and retreatment groups, or Pre, 1M, and 3M—using Fisher’s exact test. The relationships between groups were compared using a *t*-test. Receiver operating characteristic (ROC) curves were created by plotting the target values for each group, and the cutoff values were calculated using the Youden index. The TFR rate was analyzed using the Kaplan–Meier method and compared using the log-rank test. Statistical significance was set at *p* < 0.05.

## 3. Results

### 3.1. Analysis of Peripheral Blood in CML Patients Revealed That Treg Cells in the TFR Group Increased 1 Month after Discontinuation of Imatinib

We conducted an imatinib discontinuation trial in patients with CML chronic phase (CP) who were treated with imatinib for at least 3 years and had a DMR of at least 2 years (JALSG-STIM213) [11]. The relapse criterion was the loss of major molecular response (MMR), and patients who relapsed were retreated with imatinib. We collected peripheral blood samples before stopping imatinib (hereafter referred to as pre-stopping), and at 1 and 3 months after stopping imatinib and performed immunological analysis (Figure 1A). We performed a comparative analysis between two groups of patients: those who maintained TFR and those who lost MMR and received imatinib again after 3 years of trying TFR. On comparing the patient characteristics between the two groups, no difference was observed except in the IS-PCR values between undetectable and detectable minimal residual disease (MRD) at pre-stopping, as reported previously [11] (Figure 1B). We compared the proportions of T, B, NK, and myeloid-derived suppressor cells (MDSCs) using mass cytometry between the TFR and the retreatment groups, and the results showed no difference in cells of those two groups (Figure 1C and Appendix A). There was also no difference in the percentage of FoxP3^+^ CD4^+^ Treg cells in the two groups; however, there was a tendency toward an increase in the percentage at 1 month after discontinuation (Figure 1D and Appendix A).

### 3.2. Treg Variability after Discontinuation of Imatinib May Serve as a Biomarker for TFR Maintenance 

For a more detailed analysis of Treg cells or immune costimulatory molecules, we performed flow cytometry. Treg cells are known to be important in the immune response to CML [10,13,14]. Recently, we reported that eTreg cells, which have stronger inhibitory activity, also play an important role in achieving DMR in CML-CP [13], and the present TFR analysis also focused on this. Our results showed that, similar to the results obtained by mass cytometry, eTreg cells tended to increase after imatinib discontinuation in the TFR group (Figure 2A), but there was no significant difference in the eTreg, nTreg, and non-Treg cells between the TFR and retreatment groups (Figure 2B and Appendix A). However, the increase in the proportion of eTreg cells at 1 month after stopping imatinib was significantly higher than that at pre-stopping in the TFR group, whereas there was no difference between the two values in the retreatment group (Figure 2C). Interestingly, the increase in eTreg cells in the TFR group was transient and decreased again at 3 months after stopping imatinib. The calculated increasing ratio of eTreg cells from pre-stopping to 1 month after stopping imatinib was significantly higher in the TFR group than in the retreatment group (Figure 2D). When the cutoff of eTreg increasing ratio was set to 183% from the ROC curve, the TFR curve between the high increasing ratio group and the low increasing ratio group showed a significant difference (Figure 2E). These results indicate that an increase in the rate of eTreg after discontinuation of imatinib could be a biomarker for TFR maintenance. 

In terms of antitumor immune responses, an increase in eTreg cells is generally thought to decrease the antitumor immune response and favor the proliferation of tumor cells [24,29,30,31,32]. We next investigated the activation status (immunosuppressive capacity) of Treg cells using surface marker analysis. The results showed no difference in CTLA-4, one of the functional molecules for Treg suppression [33], in eTreg cells between pre-stopping and 1 and 3 months after imatinib discontinuation in the TFR group. We found no changes in either LAP or GARP expression, which are thought to be important for Treg differentiation, and no changes in Ki-67, a proliferation marker, or other immunophenotypes (Appendix A). Thus, although eTreg cells transiently increased in proportion after stopping imatinib, we did not find any activation of eTreg cells, which suggests a suppression of the antitumor immune response. In addition, transforming growth factor-β (TGF-β) has been reported to induce Treg cell differentiation [34], and there was a correlation between TGF-β and eTreg proportion during the pre-stopping phase in the TFR group, whereas there was no correlation between the two in the retreatment group (Appendix A). 

### 3.3. CD8^+^ T Cells in the TFR Group Were Activated after Discontinutaion of Imatinib

As eTreg cells increased after the imatinib discontinuation in the TFR group, we next examined the role of CD8^+^ T cells as effector cells. We found no difference in the proportion of or change in CD8^+^ T cells between the two groups (Appendix A); however, the expression of PD-1 was significantly decreased in the TFR group compared to that in the retreatment group (Figure 3A). CTLA-4 and TIGIT levels, immunosuppressive molecules similar to PD-1, tended to be lower, while 4-1BB, GITR, and ICOS, costimulatory molecules for cell activation, tended to be higher in the TFR group (Appendix A) compared to those in the retreatment group. In addition, the expression of Ki-67 was significantly higher in the TFR group, and serum interferon-γ (IFN-γ) levels were also higher in the TFR group after discontinuation of imatinib (Figure 3B,C). These data suggest that CD8^+^ T cells in the TFR group were activated more than those in the retreatment group. According to the ROC curve of Ki-67 positivity at 1 month after discontinuation of imatinib, the cutoff was set to 5.21%, and the TFR curve generated by combining the increasing ratio of eTreg cells and Ki-67 positivity in CD8^+^ T cells could distinguish the prognosis more clearly (Figure 3E). Thus, combining the variability of eTreg cells with a component of CD8^+^ T-cell activation was shown to enhance the prediction of TFR maintenance after imatinib discontinuation. In addition, we examined intracellular signaling molecules using mass cytometry. Hierarchical cluster analysis at 1 month and 3 months after discontinuation of imatinib showed that multiple molecules involved in T-cell activation tended to correlate with prognosis (Appendix A). Interestingly, in addition to activation signals such as PI3K and MAPK, SHP2, an inhibitory signaling molecule, was suggested to be associated with prognosis.

We also examined the subpopulations of CD8^+^ T cells demonstrated by CD45RA and CCR7 and found no difference between the TFR and retreatment groups in the naïve (CD45RA^+^CCR7^+^), central memory (CM; CD45RA^−^CCR7^+^), effector memory (EM; CD45RA^−^CCR7^−^), and CD45RA^+^ EM (EMRA; CD45RA^+^CCR7^−^) fractions between the TFR and retreatment groups (Appendix A). The expression of PD-1 in the CM and EM groups was significantly lower in the TFR group than in the retreatment group (Appendix A). These results suggest that an antitumor immune response was induced in the TFR group because the CM and EM fractions functioned as effectors of T cells.

These results suggested that CD8^+^ T cells were activated after the discontinuation of imatinib in the TFR group, whereas the increase in transient Treg did not reduce the immune responses in the TFR group.

### 3.4. High Imatinib Concentrations Suppress the Immune Response during the TFR Phase

Next, we focused on the blood concentration of imatinib during imatinib treatment. Before discontinuing imatinib, the mean trough concentration was 972.2 ng/mL, and there was no significant difference between the TFR and retreatment groups (Figure 4A and Appendix A). Regarding the relationship between imatinib trough concentration and Treg, we found that eTreg values after stopping imatinib were negatively correlated with imatinib trough concentration (*R*^2^ > 0.04) and showed no correlation with pre-stopping imatinib (Figure 4B). As shown in Section 3.2, eTreg increased after the discontinuation of imatinib in the TFR group. However, eTreg cells of the patients with high trough concentrations remain at low levels after discontinuation of imatinib, suggesting that high imatinib concentration is inversely correlated with changes in eTreg cells and the activation of CD8^+^ T cells after discontinuation of imatinib, especially in the TFR group (Figure 4B and Appendix A). However, there was no clear correlation between PD-1^+^CD8^+^ T-cell counts and trough values in the TFR group before and after discontinuation of imatinib, whereas there was a positive correlation after the discontinuation of imatinib in the retreatment group (Figure 4C). These results suggest that a high imatinib trough concentration may induce exhaustion of effector cells in the retreatment group. In conclusion, a history of exposure to high imatinib concentrations may have a negative effect on the maintenance of TFR. 

### 3.5. In Vitro Exposure to High Imatinib Concentrations Suppresses Treg and CD8^+^ T Cells

Lastly, we investigated the in vitro immune response after imatinib administration was discontinued. PBMCs from healthy individuals were cultured in the presence of imatinib for 3 days, after which imatinib was washed out, T cells were stimulated with anti-CD3 antibody, and Treg and CD8^+^ T cells were measured sequentially. Our results showed that imatinib suppressed Treg proliferation in a concentration-dependent manner, whereas CD8^+^ T cells showed little change (Figure 5A). However, when the positivity of Ki-67 in both cells was checked, Treg and CD8^+^ T cells were significantly reduced after exposure to 40 µM imatinib compared to lower concentrations of imatinib (Figure 5B). In other words, the number of proliferative cells decreased in a concentration-dependent manner in Treg cells, whereas the number of proliferative cells in CD8^+^ T cells decreased only at a higher imatinib concentration of 40 µM (Figure 5B). In CD8^+^ T cells, the expression of PD-1 was not observed after exposure to 40 µM imatinib but tended to slightly increase at 2.5 µM and 10 µM imatinib (Appendix A). These results are consistent with those of a clinical trial that showed no change in CD8^+^ T-cell numbers after discontinuation of imatinib, but a decrease in CD8^+^ T-cell function. Since eTreg cells were shown to be highly sensitive to imatinib, confirming that the percentage of eTreg cells at pre-stopping and after stopping imatinib administration may reflect the remaining immunocompetence of the host. 

## 4. Discussion

In this study, because most molecular recurrences occurred within 6 months of the discontinuation of imatinib, we focused on the changes in host immune status in the early TFR phase and sequentially analyzed the relationship between TFR and host immune status after imatinib discontinuation in patients with CML-CP. A month after the discontinuation of imatinib, there was a significant increase in eTreg cells in the TFR group but not in the retreatment group. An increase in eTreg cells in the TFR group is thought to be a contrasting immunological response in terms of maintaining remission because it usually decreases the antitumor immune response [24,29,30,31,32]. However, this analysis also showed the activation of CD8^+^ T cells in the TFR group compared to that in the retreatment group, suggesting that antitumor immunity was enhanced. In addition to the activation of CD8^+^ T cells, serum IFN-γ levels were significantly higher in the TFR group than in the retreated group, suggesting that the immune response was not suppressed by transient elevation in Treg cell numbers. Although the number of Treg cells, which was suppressed by imatinib for a long time, temporarily increased after imatinib discontinuation, the immune response by conventional T cells was also activated. The overall effect was thought to be an enhancement of the antitumor immune response.

It is known that the balance between CD8^+^ T cells and Treg cells is important in the antitumor immune response and that an increase in CD8^+^ T cells or a decrease in Treg cells enhances the antitumor immune response [29,35,36]. The population of eTreg/CD8 in this analysis showed a similar trend to that of eTreg cells because CD8^+^ T cells showed little change (Appendix A), and the ratio increased 1 month after discontinuation of imatinib in the TFR group, whereas no change was observed in the retreatment group (Appendix A). Therefore, we next investigated the function of eTreg and CD8^+^ T cells by evaluating the expression of their surface molecules. The expression of eTreg surface did not differ between the TFR and retreatment groups, suggesting no obvious difference in the immunophenotype of eTreg cells between the two groups. In contrast, for CD8^+^ T cells, the expression of inhibitory costimulatory molecules such as PD-1 was significantly decreased in the TFR group compared to that in the retreatment group, whereas the expression of promotive costimulatory molecules such as 4-1BB tended to increase, and a higher percentage of CD8^+^ T cells had a Ki-67-positive proliferative potential. These results indicate that CD8^+^ T cells in the TFR group were more activated than those in the retreatment group, and this difference between the two groups was observed before imatinib was discontinued.

In the analysis of antitumor immunity, an increase in the percentage of CD8^+^ T cells is generally treated as an increase in the immune response. This is because an increase in CD8^+^ T cells is expected to result in an increase in tumor-specific CD8^+^ T cells [12,27,37]. However, the results of this study showed that CD8^+^ T-cell activation was not reflected in the number of CD8^+^ T cells. In vitro analysis also showed that the percentage of CD8^+^ T cells after imatinib treatment did not change regardless of imatinib concentration, whereas the number of proliferative marker-positive CD8^+^ T cells increased significantly only under high imatinib concentrations. Therefore, it is possible that the immune status cannot be accurately assessed by simply estimating the increasing or decreasing numbers of CD8^+^ T cells or Treg cells, and a comprehensive judgment, including analysis using costimulatory molecules and proliferation markers, is necessary to evaluate the immune capacity of the host. The increase in the percentage of activated CD8^+^ T cells in the TFR group revealed by this analysis suggests that the antitumor immune response by CD8^+^ T cells may have maintained the TFR. Conversely, in the retreatment group with a high percentage of exhausted CD8^+^ T cells, the failure to elicit a sufficient antitumor immune response may have led to relapse. Interestingly, in the TFR group, Treg cells showed transient changes after discontinuation of imatinib, whereas the activation of CD8^+^ T cells was persistently observed from pre-stopping to 3 months after discontinuation of imatinib administration. Examination of intracellular signals suggested that the T-cell receptor (TCR) signaling pathway mediated by lymphocyte-specific protein tyrosine kinase (LCK) and the downstream Akt pathway were activated even at 3 months after stopping imatinib in the TFR group compared with those in the retreatment group (Appendix A). Thus, the continued activation of CD8^+^ T cells after discontinuation of imatinib may effectively maintain TFR for a long time. 

Our previous report [13] and the present results indicate that Treg cells are highly sensitive to imatinib and likely affected by imatinib discontinuation. The biomarkers for TFR are known to be the duration of TKI treatment, the duration of DMR, and the depth of remission, all of which are related to the persistence of leukemia stem cells, i.e., whether the MRD was eradicated or not. It is easy to understand that relapse is caused by MRD; however, in TFR, there is a condition called “fluctuate” in which residual *BCR–ABL1*-positive leukemia cells are observed but no apparent relapse is observed. The possibility of attaining equilibrium or elimination of residual tumor cells by immune cells has been speculated. Several reports on TFR analysis have focused on immune cells and have shown prognostic stratification by NK cell count, Treg cells, and MDSC ratio before the discontinuation of imatinib [16,17,18,19,20,21], but we could not find any relationship between them in this analysis. There was no difference in the absolute value of Treg cells between the TFR and retreatment groups in our analysis. A noteworthy point of this analysis is the discovery of differences in the variability of Treg cells after stopping imatinib, which can only be shown by sampling the same patients sequentially. Although the possibility that immune cells may be affected by the discontinuation of long-term TKIs and show different behavior has only been reported in a few other TKI discontinuation studies [10], the Treg changes observed in this study are unique in that they occurred as early as 1 month after stopping imatinib and were transient. Although Treg cells play a very important role in the immune system, they form a very small population, approximately 0.5–5% of the CD4^+^ T cells. Treg values vary widely among individuals, and it is difficult to obtain a difference solely on the basis of these values. Therefore, measuring the variability of Treg cells is considered a reasonable method for standardizing Treg values, which vary among individuals, and for its use as a biomarker. The present analysis suggests that the activation of the immune state is important for the maintenance of TFR, and that this activation state, which is not shown by the number of CD8^+^ T cells, may be inferred by changes in Treg cells, which are highly sensitive to imatinib. However, if there is no change in Treg cells after the discontinuation of imatinib, it is possible that the entire immune system has become exhausted such that changes in Treg cells could not occur. In fact, the accuracy of prognostic prediction could be improved by combining both Treg and CD8^+^ T cells (Figure 3E). 

In the TKI discontinuation study, the DADI study reported that a low number of Treg cells could be a biomarker [18]. In such studies, Treg cells were defined as CD4^+^CD25^+^CD127^low^, but this population also included FoxP3-negative non-Treg cells [22], which may have included effector cells. In this analysis, CD45RA and FoxP3 were used to classify Treg cells more accurately. We showed that the Treg cells strictly defined were highly sensitive to imatinib and, therefore, could be an indicator of the immunological reserve of the patients. Some reports have shown that NK cells are biomarkers for TFR [16,17,18,19]. Ilander et al. found a difference in CD56^bright^ cells [16], while Rea et al. found a difference in CD56^+^ or CD56^dim^ cells and no difference in CD56^bright^ cells [17]. Irani et al. also showed a difference between the two cohorts in CD56^dim^ cells [19]. Thus, although a high number of NK cells has been suggested to be associated with a favorable prognosis, it has not been established as a predictive marker. In this analysis, no difference was observed in the CD56^+^/CD16^+^ NK cells. 

In general, higher blood levels of imatinib are associated with a deeper response in the treatment of CML. However, the results of this analysis suggest that high imatinib concentrations may exhaust the immune system, which may increase the relapse rate after discontinuation of imatinib. In other words, imatinib trough concentrations that are high enough to alter immuno-kinetics may be negatively correlated with TFR maintenance in some cases. In an imatinib discontinuation study in children (JPLSG STKI-14), it was reported that only low imatinib concentrations were significantly correlated with TFR in univariate analysis [38]. The mechanism in this report has not been clarified, but it is known that high concentrations of imatinib suppress the immune system [39,40,41,42], suggesting that the immune system remains exhausted after exposure to high concentrations of imatinib, even when imatinib is discontinued. In the case of imatinib-sensitive individuals, while high imatinib exposure may eradicate residual *BCR–ABL1*-positive cells, it may also exhaust the antitumor immune response, which may not be relieved even after discontinuation of imatinib, leading to relapse. Overall, long-term exposure to high concentrations of imatinib may require caution in patients who are considering the future challenges of TFR in the future.

## 5. Conclusions

We demonstrated that the immune assessment of patients with CML in an imatinib discontinuation trial can reveal the disease prognosis after discontinuation of imatinib.

We first analyzed the immune profiles of 68 patients in an imatinib discontinuation trial. Immune conditions show large differences among individuals, especially in TKI-treated CML patients; therefore, we focused on the changes in immune status in these patients. Sequential sampling from the same patient before and after stopping imatinib revealed an association between host immune status influenced by imatinib exposure and maintenance of TFR. In the TFR group, eTreg cells were elevated 1 month after stopping imatinib, and CD8^+^ T cells were activated before and after stopping imatinib. In contrast, CD8^+^ T cells were exhausted before and after stopping imatinib administration in the retreatment group. We next confirmed that exposure to high doses of imatinib remarkably reduced the proliferation of CD8^+^ T cells.

In conclusion, optimal imatinib exposure may lead to appropriate immune responses, whereas excessive imatinib exposure may induce immune exhaustion. eTreg cells are the most sensitive cells for imatinib and may serve as biomarkers for the host immune condition and for subsequent TFR after imatinib discontinuation.

## Figures and Tables

**Figure 1 cancers-13-05904-f001:**
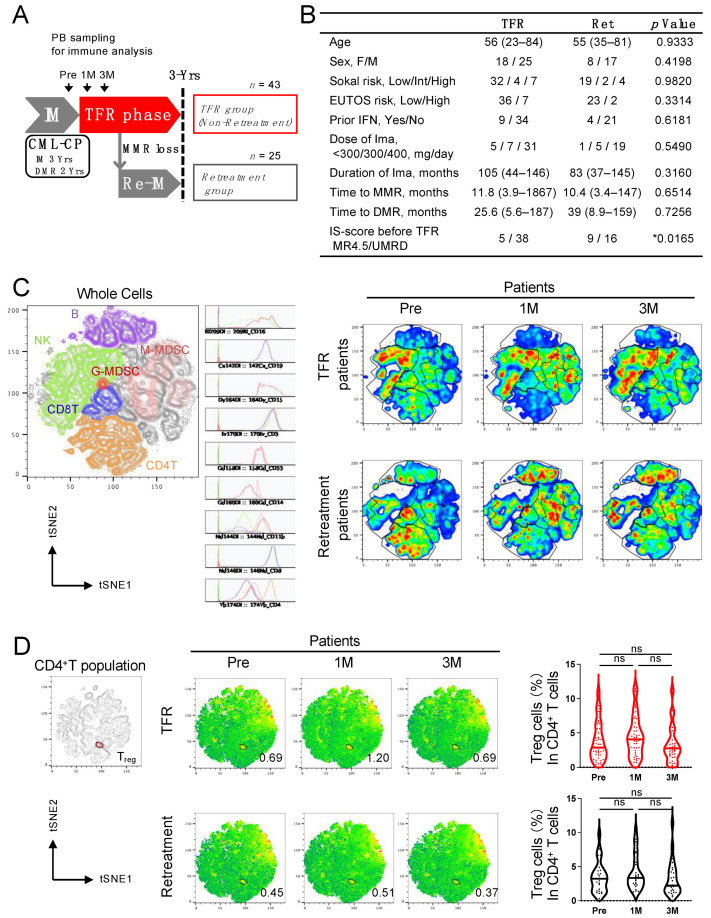
Analyses of peripheral blood (PB) immune cells in an imatinib discontinuation trial showed distinctive changes in regulatory T (Treg) cell concentrations between treatment-free remission (TFR) and retreatment groups. (**A**) Schematic summary of the JALSG STIM213 trial. Patients with chronic myeloid leukemia chronic phase (CML-CP) who received imatinib treatment for at least 3 years and who sustained deep molecular remission (DMR) for at least 2 years were eligible. MR4.5 (*BCR–ABL1*^IS^ < 0.0032%) was confirmed before stopping imatinib. TFR was defined as no findings of loss of major molecular response (MMR, *BCR–ABL1*^IS^ > 0.1%) after discontinuation of treatment. Retreatment was started immediately at the loss of MMR without a re-examination of IS-PCR. PB was sampled before stopping imatinib treatment (hereafter referred to as “pre-stopping”) and at 1 month (1M) and 3 months (3M) after stopping imatinib and was subjected to immune analyses. IM, imatinib. UMRD, undetectable molecular residual disease. (**B**) Clinical characteristics of patients in TFR and retreatment (Ret) groups. Data are presented as the median (range): age, duration of imatinib treatment, and time to MMR and DMR. Other patient characteristics are expressed in patient numbers. Fisher’s exact test was used for statistical analysis. * *p* < 0.05. (**C**) Representative *t*-distributed stochastic neighbor embedding (*t*-SNE) plots in lymphocytes (left). Histograms show the expressions of CD16, CD19, CD15, CD3, CD33, CD14, CD11b, CD8, and CD4 among fractions. Sequential *t*-SNE plots of the patient from treatment-free remission (TFR) and retreatment groups at Pre, 1M, and 3M after stopping imatinib (right). (**D**) Representative *t*-SNE plots of CD4^+^ T cells (left) and summary (right) of the frequency of FoxP3^+^CD4^+^ Treg cells. The numbers in *t*-SNE plots denote the population of FoxP3^+^CD4^+^ Treg cells. Samples obtained from the patients of TFR (upper) or retreatment (lower) groups were frozen for storage and thawed for carrying out mass cytometry. ns, not significant.

**Figure 2 cancers-13-05904-f002:**
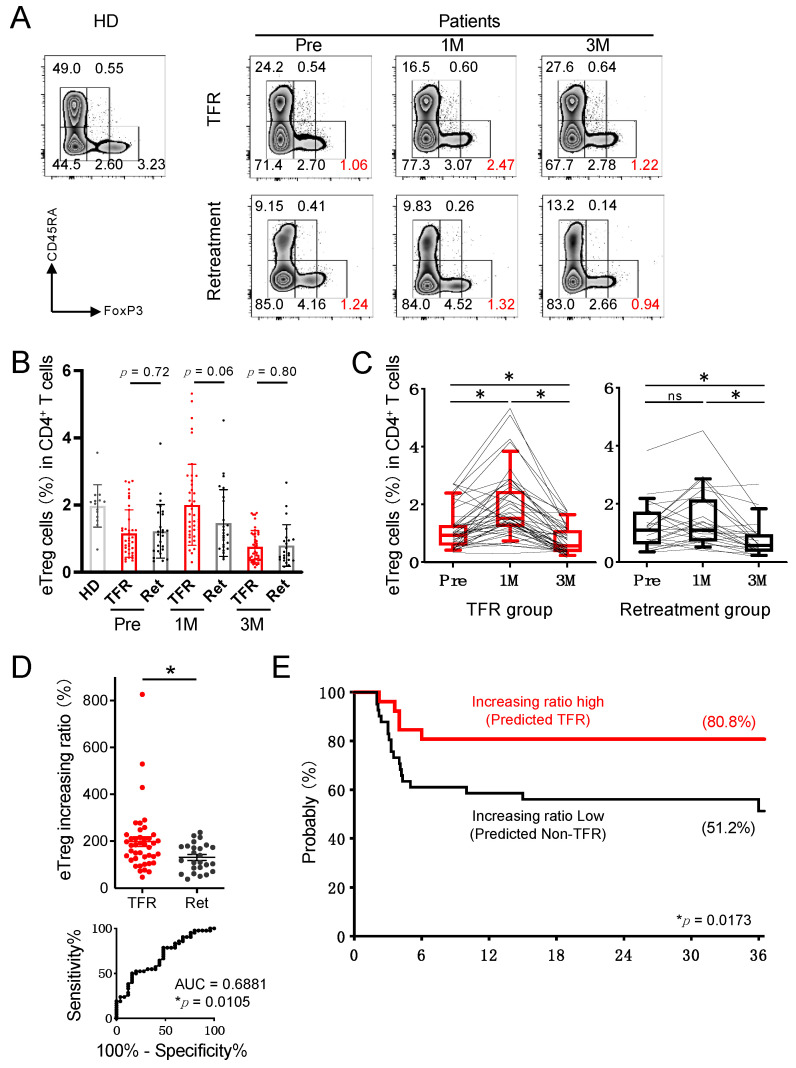
Frequency of the effector Treg (eTreg) cells in the TFR group was increased after the discontinuation of imatinib. (**A**) Representative plots of CD4^+^ T cells in a healthy donor (HD) and in CML patients from TFR (upper) and retreatment (lower) groups at pre-stopping and at 1 and 3 months after stopping imatinib (Pre, 1M, and 3M). The numbers plotted in red show the population of the eTreg fraction. (**B**) Frequencies of CD45RA^−^FoxP3^high^CD4^+^ eTreg cells among CD4^+^ T cells in HD and in patients with CML from TFR (red) and retreatment (Ret) (black) groups. (**C**) Kinetics of eTreg cells in CML patients from TFR and retreatment groups. (**D**) Scatter plot of eTreg increasing ratio from pre-stopping to 1 month after stopping (top) and ROC curve for eTreg increasing ratio (bottom). (**E**) Kaplan–Meier curve for TFR rate validated by eTreg increasing ratio. The cutoff value of eTreg increasing ratio was calculated using the Youden index. A total of 26 patients were predicted to be TFR, while 42 patients were predicted to be non-TFR. * *p* < 0.05; ** *p* < 0.001; ns, not significant.

**Figure 3 cancers-13-05904-f003:**
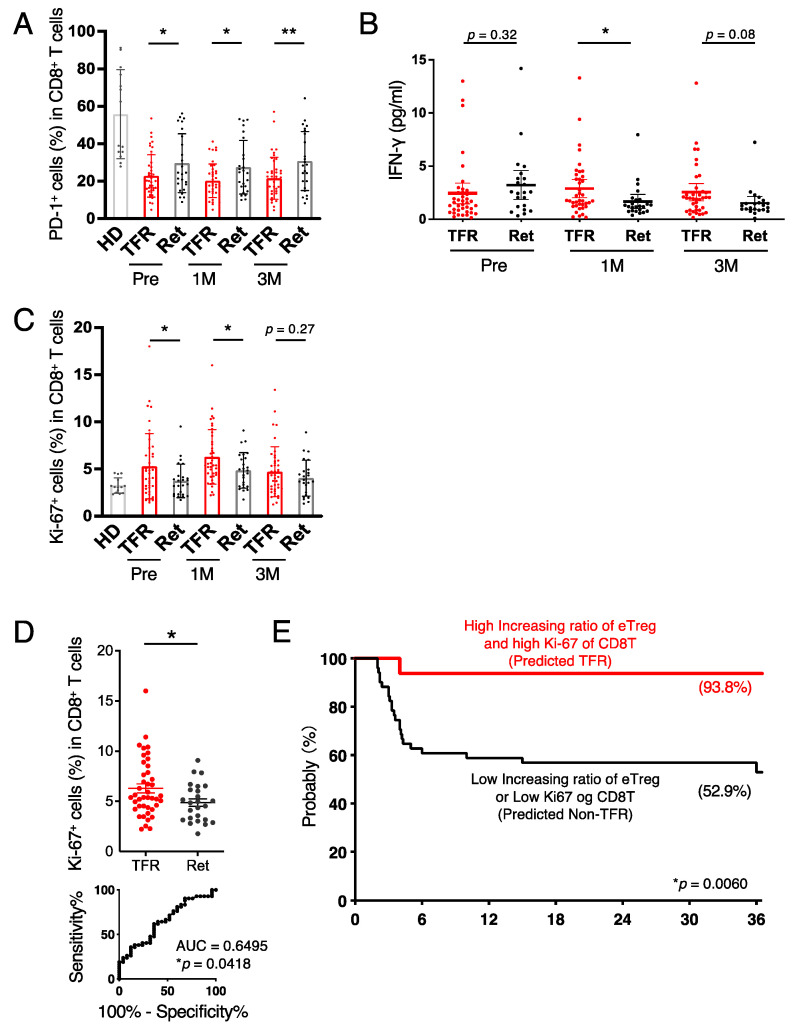
CD8^+^ T cells in the retreatment group were exhausted. (**A**) Frequencies of PD-1^+^CD8^+^ T cells in an HD and in patients with CML from the TFR (red) and retreatment (Ret) (black) groups. (**B**) Serum levels of interferon (IFN)-γ in CML patients from TFR (red) and Ret (black) groups. (**C**) Frequencies of Ki-67^+^CD8^+^ T cells in an HD and in patients with CML from TFR (red) and Ret (black) groups. (**D**) Scatter plot (top) and ROC curve (bottom) of Ki-67^+^CD8^+^ T cells at 1 month after stopping imatinib. (**E**) Kaplan–Meier curve for TFR rate validated by eTreg increasing ratio and Ki-67^+^CD8^+^ T cells at 1 month after stopping imatinib. The red line shows both a high increasing ratio of eTreg and Ki-67 expression in CD8^+^ T cells. The cutoff value of eTreg increasing ratio and Ki-67 expression in CD8^+^ T cells was calculated using the Youden index. A total of 16 patients were predicted to be TFR, while 52 patients were predicted to be non-TFR. * *p* < 0.05, ** *p* < 0.001.

**Figure 4 cancers-13-05904-f004:**
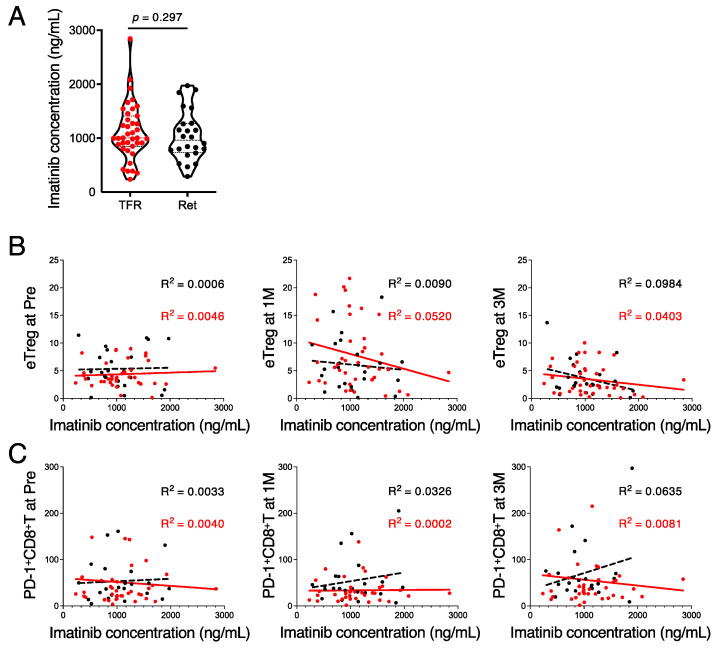
Imatinib trough concentration was inversely correlated with the frequency of Treg cells after discontinuation of imatinib. (**A**) Plasma concentrations of imatinib obtained from patients with CML in the TFR (red) and retreatment (Ret) (black) groups were measured using liquid chromatography before stopping imatinib administration. (**B**) Correlation between the frequency of eTreg and imatinib trough concentration at Pre, 1M, and 3M. Red and black dotted lines indicate the regression lines of TFR and retreatment groups, respectively. (**C**) Correlation between the frequency of PD-1^+^ CD8^+^ T cells and imatinib trough concentration at Pre, 1M, and 3M. Red and black dotted lines indicate the regression lines of TFR and retreatment groups, respectively. *R*, correlation coefficient.

**Figure 5 cancers-13-05904-f005:**
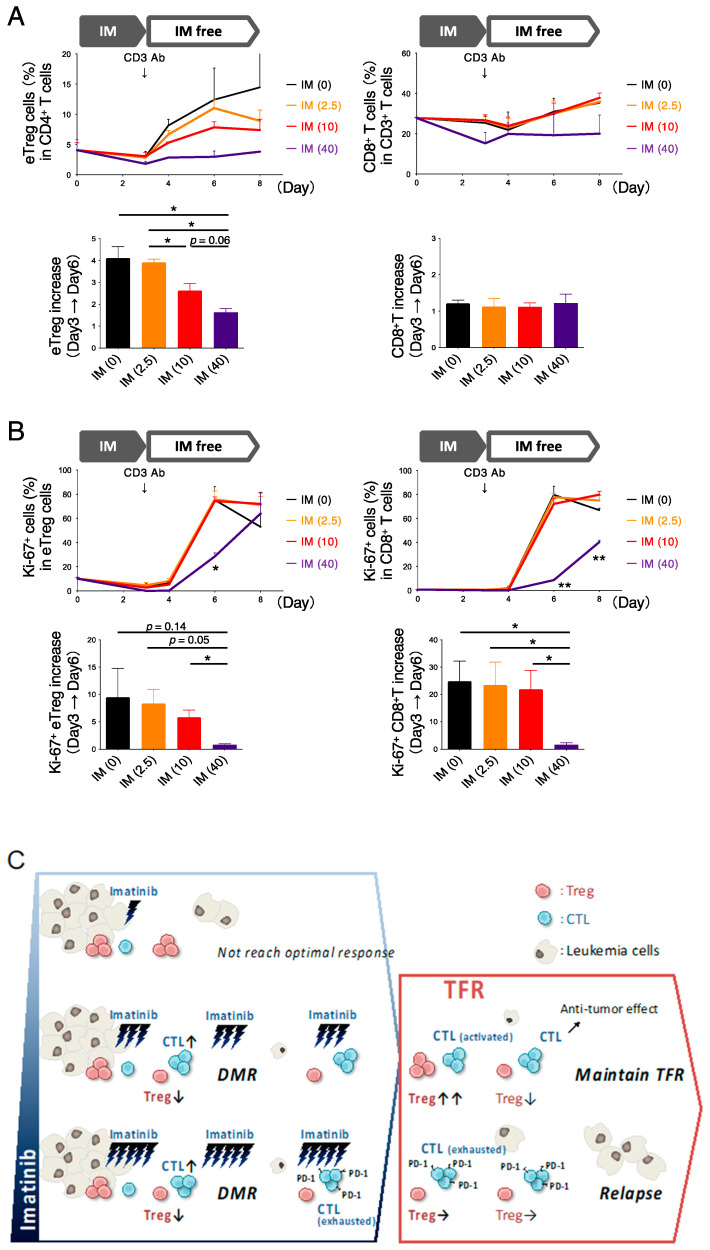
Exposure to imatinib altered the frequency of eTreg cells but not that of CD8^+^ T cells. Exposure to high concentrations of imatinib reduced not only proliferating eTreg cells but also proliferating CD8^+^ T cells. PBMCs from HDs were cultured with/without imatinib. Numbers in parentheses indicate imatinib concentration (µM). The media containing imatinib were washed out and stimulated with an anti-CD3 antibody on day 3. Cultured cells were sampled on days 0, 3, 4, 6, and 8 and subjected to flow cytometry. (**A**) Frequencies of eTreg cells (left) and CD8^+^ T cells (right). Bar graph shows the increase in eTreg cells (left) and CD8^+^ T cells (right) from day 3 to day 6. (**B**) Ki-67^+^ cells in eTreg cells (left) and CD8^+^ T cells (right). The bar graph shows the increase in Ki-67^+^ eTreg cells (left) and Ki-67^+^CD8^+^ T cells (right) from day 3 to day 6. (**C**) Schematic representation of the kinetics of regulatory (Treg) cells or CD8^+^ T cells under imatinib treatment and after stopping imatinib treatment. Lower doses of imatinib cannot achieve deep molecular remission (DMR) because Treg cells did not decrease sufficiently [13] and inhibition by BCR-ABL tyrosine kinase was insufficient. In contrast, high doses of imatinib cannot maintain treatment-free remission (TFR) because it exhausted cytotoxic T lymphocytes (CTLs). IM, imatinib; * *p* < 0.05; ** *p* < 0.001.

## Data Availability

All relevant data are included in the manuscript and its associated files.

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
