# Peer review of "Regulatory T Cell as a Biomarker of Treatment-Free Remission in Patients with Chronic Myeloid Leukemia"

_cancers, 2021, doi:10.3390/cancers13235904_

Round 1
Reviewer 1 Report
In this study, Fujioka et al. have investigated T regulatory cell (Treg) perturbation and activation after imatinib discontinuation in chronic myeloid leukemia (CML) patients. Despite the availability of a lot of data from mass cytometry, multiplexed antibody-based cytokine detection, and clinical setting, results should be better presented, organized, and discussed.
- Graphical abstract is difficult to follow.
- Some information on T cell subsets analyzed in this study should be provided in the introduction.
- Clinical characteristics of patients is missing and should be provided as a main Table. It should include also imatinib dosage, number of relapsed patients, time-to-relapse, second-line therapy, and response after second-line (all patients responded after imatinib rechallenge?). Probably no differences were described between TFR and retreated group just because retreated patients were responsive to therapy. The Authors might focus on differences between TFR and relapsed patients not-responder to imatinib. Anticoagulant used for blood drawing should be stated (heparin? EDTA? Why for cytokine detection was used serum and not heparinized plasma?).
- Catalog number of reagents should be removed.
- There are several questions related to flow cytometry protocols used. (i) Total number of cells for staining should be stated. (ii) On line 125, antibodies are used at a very low dilution (1:50) for flow cytometry: was a titration performed? Please specify and add data as supplemental. (iii) In supplemental Table 1, phospho-antibodies are present; however, no mention in materials and methods section about phospho-staining is reported. Please indicate if fixation and permeabilization (saponin?) were performed. Absence of differences in phosphoproteins are likely related to a basal detection: usually, a short-course stimulation is performed in phosFlow (please see Spurgeon BE, Aburima A, Oberprieler NG, Taskén K, Naseem KM. Multiplexed phosphospecific flow cytometry enables large-scale signaling profiling and drug screening in blood platelets. J Thromb Haemost. 2014 Oct;12(10):1733-43; Schulz KR, Danna EA, Krutzik PO, Nolan GP. Single-cell phospho-protein analysis by flow cytometry. Curr Protoc Immunol. 2012 Feb;Chapter 8:Unit 8.17.1-20; Tsai WL, Vian L, Giudice V, Kieltyka J, Liu C, Fonseca V, Gazaniga N, Gao S, Kajigaya S, Young NS, Biancotto A, Gadina M. High throughput pSTAT signaling profiling by fluorescent cell barcoding and computational analysis. J Immunol Methods. 2020 Feb;477:112667). Gating strategy and boundaries should be outlined and shown at least in supplemental. (iv) How PMT voltage setting was carried out (single-color controls? Beads?). (v) Configuration of BD LSRFortessa should be indicated (lasers?).
- On line 134, the Authors stated that values outside the range were not included in the analysis. Usually, sample dilution is performed to bring back values in the kit’s range. Was the background removed? Which negative control was used?
- Cell culture paragraph should be better detailed. How imatinib concentrations were chosen? An IC50 experiment was previously performed? How data were normalized?
- Timepoints of sampling should be better described.
- No raw data or p values are reported in the text, while the results section is mixed with discussion. Please better report data in the Results section without speculate too much as some discussion points are not really supported by results.
- UMRD are present in the TFR group. Why those patients stopped imatinib?
- From paragraph 3.2, it seems that mass cytometry does not add information compared to standard flow cytometry. Therefore, why use a very expensive and time- and labor-consuming methodology?
- Please define the eTreg ratio. Did the Authors try to calculate an eTreg/CTL ratio?
- Paragraph 3.3 describes data that are reported only in supplemental thus making difficult the reading of the manuscript.
- Supplemental Figure 1 seems more informative than main Figure 1.
- Make sure R2 and p values are correctly reported in the text and figures.
- Discussion should be shortened and should focus more on reported results.
- Several abbreviations are missing.
Reviewer 2 Report
Fujioka et al presented data regarding immunological parameters that could predict results in CML patients discontinuing treatment. The topic is of interest because around 50% of patients that discontinue treatment would eventually loss response.
Authors made a big effort to analyze an important number of immunological parameters. Data presented comes from a prospective clnical trial, which is very relevant because many of previous manuscript adressing this topic camen from retrospective studies or longitudinal studies.
The main limitation of the study is that the their main finding did not showed robust significance to explain the role of immune response in treatment discontinuation probabilities
We would like to provide some comments to take into considerations that we hope could improve the quality of the manuscript:
RESULTS:
- 3.1:
- Table 1 shown in figure 1B shows how molecular information before discontinuation is missing in around 20% of patients before. Can authors explain reasons for missing information? Can authors also provide information regarding the percentage of samples analyzed for immunological parameters in every time point?
- The authors conclude that there is a notable increase in the percentage at 1 month of FoxP3, CD4. In our opinion data shown in figure 1C shows a tendency without statistical significance. We suggest to modify the sentence.
- Can authors provide information of the BCR-ABL levels in every time point and study the correlation between the FOXP3 CD4 and an increasing of BCR-ABL levels?
- 2.Treg Variability After Discontinuation of Imatinib May Serve as a Biomarker for TFR Maintenance:
- The majority of information from lines 189 to 199 should be mentioned in methods and not in results section
- Authors conclude that when the cut-off of eTreg an increasing ratio of183% could predict probabilities of TFR. However, it seams that there are very few patients in the group of ratio above 180%, could authors comment?
- 3. Authors showed how combining the variability of eTreg cells with a component of CD8+ T-cell (figure 3 E) activation could better predict outcomes after treatmend discontinuations. Again, can authors provide information regarding to total number of patients that are present in both groups?
DISCUSSION:
The authors nicely conclude their results, however, we are missing a discussion comparing their results with previous results found by others manuscripts commented in the introduction.
Could authors mention potential mechanisms to improve outcomes in TFR strategies based in their results?
Round 2
Reviewer 1 Report
The Authors have addressed all raised comments.
Reviewer 2 Report
I would like to thank the authors for includding our suggestions. We believe they did a great job and the manuscript has considerable improved.
I believe is now suitable for publicacion